# LLM-informed Object Search in Partially-Known Environments via Model-based Planning and Prompt Selection

## Abstract

We present a novel LLM-informed model-based planning framework for object search in partially-known environments. Our approach uses an LLM to estimate statistics about the likelihood of finding the target object when searching various locations throughout the scene that, combined with travel costs extracted from the environment map, are used to instantiate a model, thus using the LLM to inform, rather than replace, planning and achieve effective search performance. Moreover, the abstraction upon which our approach relies is amenable to deployment-time model selection via the recent *offline replay* approach, an insight we leverage to enable fast prompt and LLM selection during deployment. Simulation experiments demonstrate that our LLM-informed model-based planning approach outperforms the baseline planning strategy that fully relies on LLM and optimistic strategy with as much as 11.8% and 39.2% improvements respectively, and our bandit-like selection approach enables quick selection of best prompts and LLMs resulting in 6.5% lower average cost and 33.8% lower average cumulative regret over baseline UCB bandit selection. Real-robot experiments in household settings demonstrate similar improvements and so further validate our approach.

## 1 Introduction

We consider the problem of *object search* in partially-known household environments, in which a robot is tasked to find an object of interest and can use large language models (LLMs) to inform the robot's behavior. Effective object search in these scenarios often requires (i) considering the impacts of the robot's immediate actions far into the future and (ii) an ability for the robot to continuously self-evaluate during deployment to improve itself over time.

Model-based planning with a high-level action abstraction is a common decision-making framework for embodied intelligence, since it not only enables deciding what action the robot should do next but also enables reasoning farther into the future about the long-term value~~goodness~~ of a particular course of action. While many existing approaches that leverage LLMs for object search tasks use a high-level action abstraction in which the robot's actions corresponds to exploration of spaces that might contain the target object (Dorbala et al., 2023; Zhou et al., 2023; Yu et al., 2023; Arjun et al., 2024; Ge et al., 2024; Rajvanshi et al., 2024), they often prompt the LLMs to directly decide what action the robot should pick next (Zhou et al., 2023; Dorbala et al., 2023; Yu et al., 2023) without using a plannning framework. As such, LLMs can struggle to achieve good performance on many planning tasks (Valmeekam et al., 2022; 2023; Kambhampati, 2024; Kambhampati et al., 2024), particularly those necessary for embodied intelligence, including object search. Moreover, it is well-established that LLMs perform poorly on quantitative reasoning tasks (Lewkowycz et al., 2022; Arora et al., 2023; Boye & Moell, 2025; Rahman & Mishra, 2025), a limitation that translates to difficulties in choosing the *best performing* of a family of potentially suitable actions, resulting in greedy or myopic behavior that model-based reasoning is designed to avoid. However, it is not straightforward to integrate an LLM with a model-based planner and thus unclear how to best benefit from both model-based reasoning and the useful commonsense understanding an LLM can provide.

In addition, the agent's performance on object search tasks depends strongly on the choice of prompting strategy—e.g., the prompt text, description of robot's environment, in-context examples,

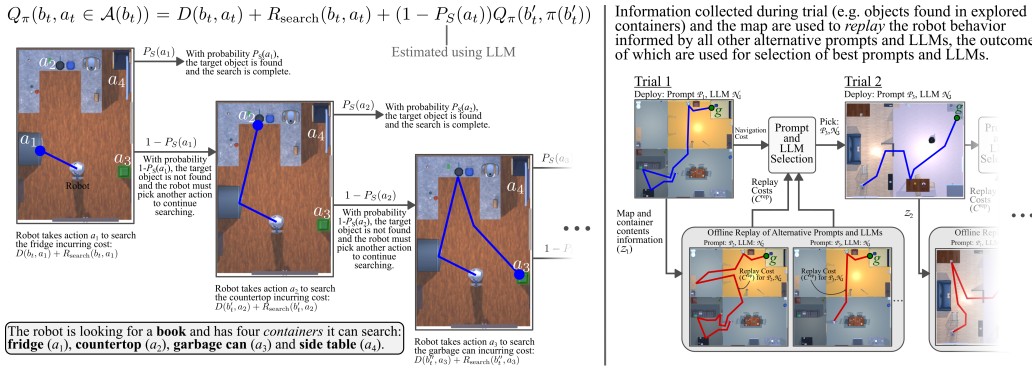

(a) LLM-informed Model-Based Planning for Object Search      (b) Deployment-time Selection of Prompts and LLMs

Figure 1: Our **high-level action abstraction**, in which actions correspond to searching available containers to look for target object, enables (a) **model-based planning** in which an **LLM informs the likelihood** $P_S$ **of finding target object** in a container and, (b) **fast deployment-time selection of prompts and LLMs** for effective object search performance. Illustration in (a) demonstrates the search problem implicit in the Bellman equation, which requires building a search tree to find the search policy that minimizes the expected cost.

etc.—and LLM model, since choosing different prompts or LLMs can result in varied performance when deployed, particularly when the deployment-time environments differ from those that were considered when designing such prompts. As such, selecting only a single prompting strategy or LLM in advance will not always elicit the best deployment-time performance. Instead, the robot should be able to choose from more than one prompting strategies or LLMs and evaluate each of those to pick the best ones during deployment. However, the process of deploying and repeatedly trying out prompting strategies or LLMs until a clear winner emerges can be problematically time consuming in general, requiring many trials to choose between them. Recent work in the space of point-goal navigation (Paudel & Stein, 2023) presents *offline alt-policy replay*, in which model-based counterfactual reasoning can be used to afford choosing the best of a family of learning-informed navigation policies, a strategy we seek to leverage for prompt and LLM selection.

To achieve effective object search performance, we therefore require a model-based approach that both informs and is informed by an LLM: with which we can plan using the commonsense world knowledge of LLMs and also introspect during deployment so as to quickly allow the system to select the best performing prompting strategy or LLM. It is a key insight of this work that a model-based planning framework for LLM-informed object search in partially-known environments can be built upon the same high-level action abstraction used in similar approaches designed for learning-informed planning under uncertainty (Stein et al., 2018). In addition, the same high-level action abstraction also affords *offline replay* (Paudel & Stein, 2023), and so can facilitate deployment-time evaluation of prompts and LLMs for object search tasks—the outcomes of which can then be used to quickly select the best performing prompts and LLMs.

In this work, we present an LLM-informed model-based planning framework for object search in partially-known environments, and an accompanying approach for deployment-time selection of the best prompts and LLMs for such LLM-guided object search tasks (Figure 1). Our model-based planning framework leverages a frequently-used high-level action abstraction where the robot's actions correspond to revealing unsearched/unopened containers or furniture to look for a target object and leverages an LLM to make predictions about statistics of uncertainty—namely, likelihood of finding an object of interest in a location—to *inform*, rather than *replace*, model-based object search. Further leveraging this abstraction, we enable *fast deployment-time selection* of prompts and LLMs, a capability unique in this domain, by leveraging the *offline replay* approach of Paudel & Stein (2023). Our contributions are as follows:

- We identify a high-level action abstraction as a key enabler of both *model-based planning* with LLMs and *introspection*, with which such LLM-based systems can self-evaluate deployment-time behaviors.

- We present a novel approach for LLM-informed model-based high-level planning for object search in partially-known environments that integrates predictions about uncertainty from LLMs and known traversal costs from the occupancy map.

- Leveraging the action abstraction upon which planning relies, we demonstrate fast bandit-like selection of best prompts and LLMs from a family of candidate prompts and LLMs used for guiding the robot behavior in object search tasks.

Experiments in simulated ProcTHOR environments demonstrate that our LLM-informed model-based planning framework for object search outperforms LLM-based baselines that directly ask an LLM what action to pick and optimistic baselines, resulting up to 11.8% and 39.2% improvements respectively. In addition, our prompt selection approach enables quick selection of best prompts and LLMs from a family of prompts and LLMs, resulting in 6.5% lower average cost and 33.8% lower average cumulative regret over baseline upper confidence bound (UCB) bandit selection. Real-robot experiments with a LoCoBot robot in an apartment building show improvements for both our LLM-informed model-based planning approach and our prompt selection approach.

## 2 RELATED WORK

**LLMs and VLMs for Object Search**   Many recent works have explored the use of LLMs and vision language models (VLMs) and for object search tasks (Dorbala et al., 2023; Zhou et al., 2023; Yu et al., 2023; Arjun et al., 2024; Ge et al., 2024; Rajvanshi et al., 2024). These works use LLMs or VLMs for their commonsense world knowledge to decide where to search (Zhou et al., 2023; Dorbala et al., 2023; Yu et al., 2023). However, as they do not use a planning framework, they are often fairly myopic in their search strategies. Our work focuses on leveraging the commonsense knowledge from LLMs to *inform* rather than *replace* planning to enable reasoning about long-horizon impacts of robot's search actions: a capability important to achieve good performance for object search in partially-known environments.

**LLMs and Planning**   LLMs have been widely used as planners (Irpan et al., 2022; Rajvanshi et al., 2024; Silver et al., 2022; Song et al., 2023; Wu et al., 2024) or to augment planning (Guan et al., 2023; Hazra et al., 2024; Zhao et al., 2024; Liu et al., 2023a)(Nayak et al., 2024; Zhang et al., 2025; Ling et al., 2025). LLMs have been recently used to directly solve task planning problems in planning domain definition language (PDDL), but their abilities to generate feasible or correct solutions are brittle (Valmeekam et al., 2022; 2023). Some recent approaches therefore aim to integrate classical planning methods with LLMs (Guan et al., 2023; Liu et al., 2023a; Hazra et al., 2024)(Nayak et al., 2024; Zhang et al., 2025; Ling et al., 2025). Our work is specifically focused on extracting statistics of high-level exploratory actions from LLMs to inform a model-based planning framework.

**Prompt Selection**   Prompt selection, which falls under a broader area of prompt engineering (Sahoo et al., 2024; Liu et al., 2023b), deals with selecting the prompts that achieve the best LLM performance on downstream tasks (Yang et al., 2023). While there are approaches that aim to select the best prompts from predesigned templates (Liao et al., 2022; Liu et al., 2023b; Sorensen et al., 2022; Yang et al., 2023), these approaches focus on selecting prompts that gets the best responses from LLMs on various benchmarks and hence are not suitable for deployment-time selection of prompts in LLM-informed object search tasks, the focus of this work.

## 3 PROBLEM FORMULATION

**Object Search in Partially-Known Environments**   Our robot is tasked to find a target object $g$ in a household environment in minimum expected cost, measured in terms of distance traveled. The environment consists of rooms, containers and objects. *Containers* are entities in the environment that can contain other objects: bed, dresser, countertop, etc. The containers are located in different rooms in the household environment. The belief state $b_t = \{m_t, q_t\}$ consists of the map $m_t$—with *a priori* known locations of rooms and containers but what objects exist in the containers are not known—and the robot pose $q_t$, both at time $t$. The robot must navigate to containers and search them to look for the target object. Unexplored containers form the robot's action space $\mathcal{A}$ and the robot's policy $\pi$ maps the belief state $b_t$ to a container search action $a_t \in \mathcal{A}(b_t)$. Our search policies are informed by LLMs and so depend upon the choice of LLMs and prompts used to query the LLMs.

We presume that the robot has access to a low-level navigation planner and controller that can be used to move about and interact with the environment. As such, the aim of our planner is to determine the sequence of container search actions that minimizes the expected cost of finding the

target object. The performance of the robot during deployment is measured as the average distance traveled by the robot to find the target object across a sequence of trials, where each trial is held in a distinct map to find an object sampled uniformly at random from the environment.

**Prompt Selection**    We consider that the robot's policy has access to multiple prompt templates and LLMs each represented as $\theta = (\mathcal{P}, \mathcal{N})$ where $\mathcal{P}$ denotes prompt template and $\mathcal{N}$ denotes LLM. As such, the robot has access to a family of search policies $\Pi = \{\pi_{\theta_1}, \pi_{\theta_2}, \cdots, \pi_{\theta_N}\}$ each with a unique prompt-LLM pair. The objective of prompt selection is to pick the policy with a prompt-LLM pair $\theta$ whose corresponding search actions result in minimum expected cost of finding target objects during deployment over multiple trials in distinct partially-known environments:

$$\pi_\theta^* = \arg\min_{\pi_\theta \in \Pi} \mathbb{E}[C(\pi_\theta)] \tag{1}$$

where $\mathbb{E}[C(\pi_\theta)]$ is the expected cost incurred by the robot upon using policy $\pi_\theta$ with a prompt-LLM pair $\theta$ during deployment. This problem can be formulated as a multi-armed bandit problem (Sutton & Barto, 2018), solved via black-box selection algorithms like UCB (Lai et al., 1985) using Eq. (2):

$$\pi_\theta^{(k+1)} = \arg\min_{\pi_\theta \in \Pi} \left[ \bar{C}_k(\pi_\theta) - c\sqrt{\frac{\ln k}{n_k(\pi_\theta)}} \right] \tag{2}$$

where $\bar{C}_k(\pi_\theta)$ is the average cost over trials 1-through-$k$ in which policy $\pi_\theta$ with prompt-LLM pair $\theta$ was selected, $n_k(\pi_\theta)$ is the number of times policy $\pi_\theta$ was selected until trial $k$, and $c > 0$ is a parameter controlling the balance between exploration and exploitation. However, such approaches can be slow to converge, requiring the robot to go through multiple trials of poor performance before the best policies can be identified. White-box approaches can accelerate selection (Paudel & Stein, 2023; Paudel et al., 2024), but rely on planning abstractions that support counterfactual reasoning about robot behavior. It is our insight that LLM-informed planning strategies can be made compatible with such approaches and so can afford prompt and LLM selection in this setting.

## 4    LLM-INFORMED MODEL-BASED PLANNING FOR OBJECT SEARCH

Here, we introduce our approach for LLM-informed model-based planning for object search before we discuss prompt selection in Section 5.

We want an approach that can leverage the commonsense world knowledge of LLMs to *inform* model-based planning rather than using LLMs to replace planning altogether. To achieve this, we introduce an approach for LLM-informed model-based object search, in which we seek to perform model-based planning wherein planning is augmented by the predictions generated by an LLM about the object locations.

Our approach takes inspiration from the learning over subgoals planning (LSP) approach of Stein et al. (2018). Their approach, designed around the aim of effective long-horizon point-goal navigation, is centered around using learning to estimate statistics associated with temporally extended actions for exploration; a learned model, trained in environments similar to those the robot sees when deployed, estimates the goodness of each such exploratory action and the likelihood that exploring the space that the action corresponds to will reach the unseen goal.

Our LLM-informed model-based object search framework adopts a similar planning abstraction. In this framework, each high-level action corresponds to searching the *containers*, which are entities in the environment that contain other objects: bed, dresser, countertop, etc. A search policy $\pi$ specifies the sequence of search actions the robot intends to take to find the target object. Each such search action $a_t \in \mathcal{A}(b_t)$ has an immediate cost of first traveling to the container—corresponding to a distance $D(b_t, a_t)$ computed via A* from the occupancy grid—and then searching the container for the target object, which has a (known) search cost $R_{\text{search}}(b_t, a_t)$. With a probability $P_S$, the container contains the target object and so the corresponding search action successfully finds the object. Otherwise, with probability $1 - P_S$, searching continues in other containers after picking another container search action (Figure 1(a)). The expected cost of a search action $a_t$ under policy $\pi$ is computed using a Bellman equation:

$$Q_\pi(b_t, a_t \in \mathcal{A}(b_t)) = D(b_t, a_t) + R_{\text{search}}(b_t, a_t) + (1 - P_S(a_t))Q_\pi(b_t', \pi(b_t')) \tag{3}$$

The robot's policy $\pi(b_t) \equiv \arg\min_a Q_\pi(b_t, a \in \mathcal{A}(b_t))$ can be used to compute a search plan: the sequence of actions that minimizes the expected cost via Eq. (3) to find the target object. To speed up planning, we limit the action space using heuristics to choose up to eight containers with high likelihoods $P_S$ and low travel costs $D$ and select among them the container with lowest expected cost, incorporating additional containers as the robot moves and searches. It should be noted that our planning strategy is complete since, explored containers are removed from candidate action set and so in the worst case, the robot explores all available containers to find the target object if it exists. Using an LLM as the knowledge repository of where common objects of interest might be located in the given environment, we prompt the LLM provide an estimate of the marginal probability $P_S$ and use it to compute the expected cost via Eq. (3). Such augmentation of planning with predictions from LLMs enables effective reasoning without explicitly relying on LLMs for multi-step reasoning thereby enabling improved performance.

## 5 PROMPT SELECTION FOR LLM-INFORMED OBJECT SEARCH

### 5.1 OVERVIEW OF PROMPT SELECTION

When using an LLM to inform planning for object search, prompts used to query LLMs for object likelihood predictions would ideally result in effective performance. However, effectiveness of a plan in the context of object search in partially-known environments can only be realized after the robot executes them in the environments. Thus, a poor prompting strategy may only be identified as such after the robot deploys and relies upon that LLM and prompt combination—a costly strategy of trial-and-error for robot navigation tasks. Instead, if we could identify poor prompts while limiting the need to deploy them during deployment, we could rule them out quickly and prioritize selection of the best prompts to enable improved robot performance.

It is a key insight of this work that *offline replay* approach by Paudel & Stein (2023) can be used to select between prompts without the robot having to deploy the plans informed by LLMs using such family of prompts. While used by Paudel & Stein (2023) in the context of point-goal navigation in partially-mapped environments to replay the behavior of alternative policies without having to deploy them, we adapt offline replay to determine what the robot would have done if it had instead used a different prompt or LLM to guide its behavior. Costs from offline replay (Figure 2) of alternative prompts and LLMs, $\bar{C}_k^{\text{rep}}$ (averaged over trials 1-through-$k$) can then be used in UCB bandit-like selection strategy (Figure 1(b)) similar to that of Paudel & Stein (2023) to pick the policy $\pi_\theta$ with prompt-LLM pair $\theta$ for trial $k + 1$ as:

$$\pi_\theta^{(k+1)} = \arg\min_{\pi_\theta \in \Pi} \left[ \max \left( \bar{C}_k^{\text{rep}}(\pi_\theta), \bar{C}_k(\pi_\theta) - c\sqrt{\frac{\ln k}{n_k(\pi_\theta)}} \right) \right] \quad (4)$$

### 5.2 OBJECT SEARCH DURING A TRIAL

In each trial $k$ when the robot is deployed in a partially-known environment to look for a target object $g$, it uses Eq. (4) to choose one of the policies $\pi_\theta^{(k)} \in \Pi$ defined by its prompt-LLM pair $\theta = (\mathcal{P}, \mathcal{N})$, uses the prompt $\mathcal{P}$ to query the LLM $\mathcal{N}$ for object likelihood $P_S$ and computes the next action $a_t$ corresponding to searching one of the unexplored containers using Eq. (3). The robot searches the container corresponding to action $a_t$ to look for the target object $g$, repeating planning and search each time the target object is not found. The total distance traveled by the robot to find the object is the cost $C_k(\pi_\theta)$ for trial $k$. The robot stores the information $\mathcal{Z}_k$ about the contents of all containers it explored and the known map of the environment to be used later for offline replay (Section 5.3).

### 5.3 PROMPT SELECTION WITH OFFLINE REPLAY OF ALTERNATIVE PROMPTS AND LLMS

After a trial $k$ is complete, our robot uses policy $\pi_{\theta'}$ with alternative prompt-LLM pair $\theta'$ and computes an alternative search plan. However, deploying such alternative plans is expensive in general. Instead, we use the hindsight information $\mathcal{Z}_k$ about the location of the target object found in trial $k$ and the existing map to *replay* what the robot would have done if it had deployed a plan corresponding to an alternative prompt-LLM pair $\theta' = (\mathcal{P}', \mathcal{N}')$ (Figure 2). Since we now know in advance which container contained the target object $g$ based on the information $\mathcal{Z}_k$ and pessimistically

assume that all other containers would not have contained the target object, we can compute the cost of following a separate policy: the length of the trajectory the robot would have taken by following the alternate search policy to find that target object. The offline replay of a policy resembles actual deployment: replanning is done until the replayed policy suggests searching the container where the target object was found during deployment, accumulating traversal costs along the way. The

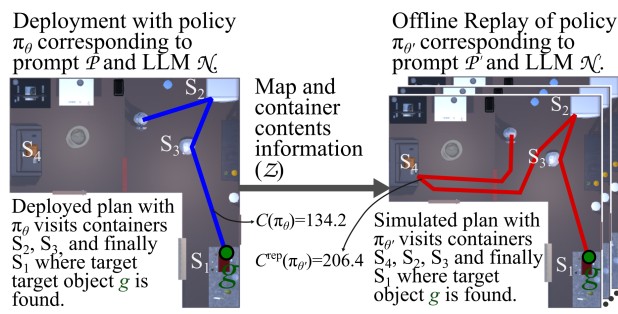

Figure 2: Overview of offline replay of alternative prompt and LLMs

average cost over trials of the offline-replayed plans, $\bar{C}_k^{\text{rep}}(\pi_{\theta'})$, for an alternative prompt-LLM pair $\theta'$ and average costs over trials of the chosen prompt-LLM pairs $\theta$ in trial $k$, $\bar{C}_k(\pi_\theta)$ are together used to pick the prompt and LLM in subsequent trials $k+1$ using Eq. (4). While the replay costs based on pessimistic assumptions may be somewhat biased, this has been empirically shown to more tightly constrain bandit-like selection compared to other relaxed assumptions in prior work of Paudel & Stein (2023) that inspired our offline replay approach.

## 6 SIMULATION EXPERIMENTS AND RESULTS

### 6.1 EXPERIMENT DESIGN

We perform simulation experiments for object search in 150 distinct household environments based on the PROCTHOR (Deitke et al., 2022) dataset, which consists of procedurally generated homes. Our robot has access to the underlying occupancy grid of the environment and what containers exist in what rooms, yet the contents of the containers are not known to the robot. Containers include any entities that could contain an object and so include more traditional containers, such as cabinets or boxes, but also surfaces, including tables, countertops, and bookshelves. The robot must travel to the container locations and search the containers to find the object of interest. For the purposes of our experiments, we do not directly simulate the robot executing manipulation skills (for example, to open a fridge or a cabinet), and we treat objects as being instantaneously revealed upon reaching a container and so assign search cost $R_{\text{search}} = 0$.

**Policies** We perform object search experiments with our LLM+MODEL model-based planner discussed in Section 4 and also design two baseline policies, LLM-DIRECT and OPTIMISTIC+GREEDY discussed below:

**LLM+MODEL** This is our LLM-informed model-based planner that uses an LLM to obtain object likelihood probabilities and then uses Eq. (3) to select the best container search action as discussed in Section 4.

**LLM-DIRECT** This LLM-informed baseline policy directly prompts the LLM to respond with the container the robot should search next, instead of asking for probabilities as we do with our LLM+MODEL planner. As such, LLM-DIRECT policy does not use a planning framework to compute actions and instead directly executes actions picked by the LLM from a list of all available container search actions.

**OPTIMISTIC+GREEDY** This non-LLM baseline optimistically assumes that all containers could contain the target object and greedily searches the nearest container, replanning until the target object is found.

**LLM Variants** We experiment with two LLMs, GPT-5 Mini and Gemini 2.5 Flash, for object search tasks in PROCTHOR household environments.

**Prompt Design** We construct multiple prompts to query the LLMs for guiding the robot behavior. The prompt design for LLM+MODEL policy and LLM-DIRECT policy are slightly different since for LLM+MODEL we want the LLM to generate probability values for each container, while for the LLM-DIRECT policy, the LLM should directly output which container the agent should search next. While such prompts might be constructed with variations in language, context and the role that LLM should

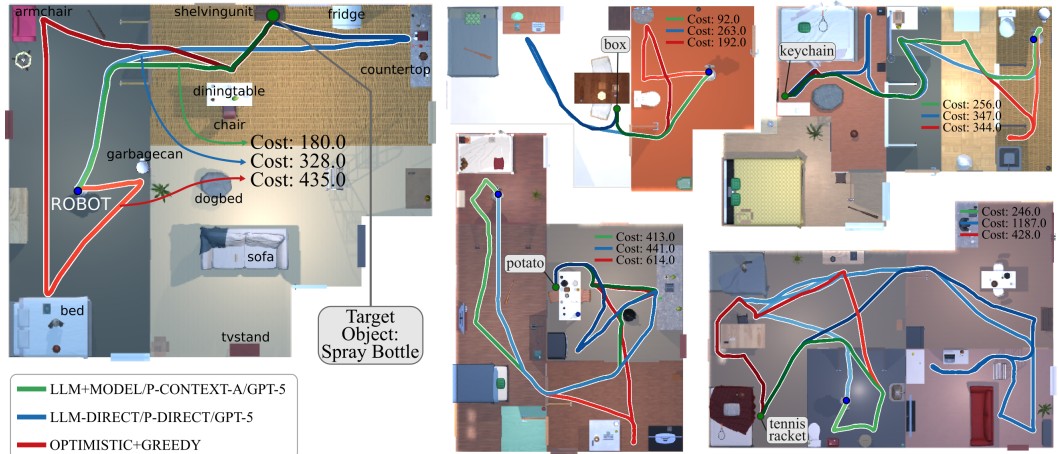

Figure 3: **Sample robot trajectories:** Using GPT-5 as LLM, our LLM+MODEL policy finds the target object with lower costs compared to LLM-DIRECT and OPTIMISTIC+GREEDY policies.

play in the interaction, each prompt includes a question asking the LLM to respond with either a probability value (for LLM+MODEL policy) or the name of the container to search (for LLM-DIRECT policy). We design three prompt templates for LLM+MODEL policy: P-CONTEXT-A, P-CONTEXT-B and P-MINIMAL, and one prompt template for LLM-DIRECT baseline policy: P-DIRECT. Examples of each are included in Appendix A.

**P-CONTEXT-A, P-CONTEXT-B:** These prompts are designed around four main elements: (i) a description of the setting and the role that the LLM will serve, (ii) a description of the house including a list of the rooms present and the containers they contain, (iii) an example for reference, and (iv) the query asking for the probability of finding the object of interest in a container within a particular room. While the semantic meaning of these prompts are similar (see Appendix A), each of these differ in terms of the language is used in the prompt text.

**P-MINIMAL:** This prompt doesn't include any of the aforementioned contexts about the LLM's role, environment description and reference example, and only includes the query asking for the probability of finding the target object in a container within a particular room.

**P-DIRECT:** This prompt for LLM-DIRECT policy is designed around five main elements: (i) a description of the setting and the role that the LLM will serve (ii) an example interaction for reference (iii) a description of the house including a list of the rooms present and the distances between them (iv) list of available containers that the robot can explore, and (v) the query asking which container the robot should explore to find the target object quickly. It should be noted that we include the distances in the prompt because we expect the LLM to behave like a planner and so provide all necessary information needed to plan effectively.

**Prompt Selection** We compare the prompt/LLM selection approach that leverages a high-level action abstraction to facilitate offline replay as discussed in Sec. 5, referred to as Replay Selection, against a baseline black-box UCB bandit selection approach. Since our selection approach is enabled by our high-level action abstraction, our other strategies that do not explicitly use planning or query LLMs—yet use the same underlying abstraction—are also amenable to offline replay, and hence can be included as a part of candidate strategies for bandit-like selection. As such, for our~~For these~~ experiments, selection seeks to choose between all nine strategies for object search from Table 1. Each deployment lasts for 100 trials, each in a distinct PROCTHOR map, over which selection proceeds. While the UCB Selection uses only the deployment cost of a strategy to pick the policy-prompt-LLM combination for subsequent trials using Eq. (2), Replay Selection additionally uses the offline replay costs of all other policy-prompt-LLM combination to pick the strategy for next trial via Eq. (4).

| Policy / Prompt / LLM | Avg. Navigation Cost |
|---|---|
| LLM+MODEL (ours) / P-CONTEXT-A / GPT-5 | **242.38** |
| LLM+MODEL (ours) / P-CONTEXT-B / GPT-5 | 252.14 |
| LLM+MODEL (ours) / P-MINIMAL / GPT-5 | 263.94 |
| LLM-DIRECT (baseline) / P-DIRECT / GPT-5 | 274.94 |
| LLM+MODEL (ours) / P-CONTEXT-A / Gemini | 224.70 |
| LLM+MODEL (ours) / P-CONTEXT-B / Gemini | **215.59** |
| LLM+MODEL (ours) / P-MINIMAL / Gemini | 266.20 |
| LLM-DIRECT (baseline) / P-DIRECT / Gemini | 233.41 |
| OPTIMISTIC+GREEDY (baseline) / – / – | 354.34 |

Table 1: **Navigation Costs for Object Search Tasks**: Average costs incurred by the robot when deploying different combinations of policy, prompt and LLM in 150 distinct maps. Bold values indicate best performers with GPT-5 and Gemini respectively.

## 6.2 POLICY PERFORMANCE RESULTS

We deploy different combinations of policies, prompts and LLMs (the state of the art GPT-5 and Gemini 2.5) each in 150 distinct maps from the PROCTHOR household environments, matching our discussion above. The average navigation costs for each such combinations are shown in Table 1. We also include example robot trajectories for three of the representative policies in Figure 3.

**Results with the GPT-5 LLM**   When using GPT-5 as the LLM to estimate object discovery likelihoods and so guide robot behavior, our model-based LLM+MODEL planners outperform the fully-LLM-based LLM-DIRECT policy, with our improvements between 4.0–11.8%, showing that our LLM-informed model-based planning approach enables considerable improvements in object search performance in partially-known environments. Additionally, both the LLM-DIRECT LLM-based baseline and our LLM+MODEL policies outperform the uninformed OPTIMISTIC+GREEDY baseline policy as they can both leverage the commonsense reasoning of the LLM to search more effectively. Our LLM+MODEL planners demonstrate improvements between 25.5–31.6% over the OPTIMISTIC+GREEDY policy.

**Results with the Gemini LLM**   Using Gemini as the LLM for guiding the robot behavior, we outperform the LLM-DIRECT policy with all our model-based LLM+MODEL planners except for the planner relying on the P-MINIMAL prompt, which includes no context about the surrounding environment; the remaining LLM+MODEL policies improve upon LLM-DIRECT baseline policy, achieving up to a 7.6% improvement. Additionally, our LLM+MODEL planners also outperform the uninformed OPTIMISTIC+GREEDY policy with improvements between 24.9–39.2%.

**Discussion**   We observe that our LLM-informed model-based planning approach outperforms those that fully relies on LLM to pick robot's search actions. These results highlight the importance of using a model-based planning framework in tandem with LLMs, rather than using LLMs in place of planners, to benefit from the strengths of both—a key focus of this work. Moreover, we observe that even the same prompts used with different LLMs can result in significantly different performances: for GPT-5, P-CONTEXT-A prompt outperforms all other strategies, and for Gemini, P-CONTEXT-B prompt outperforms all other strategies. This result highlights that relying on a single prompt or LLM may not always yield the best performance and as such, one must select the best prompt and LLM combination *during deployment* to maximize good performance in the environment the robot is deployed in.

## 6.3 PROMPT SELECTION RESULTS

As discussed in Sec. 5, it is another key insight of our work that the action abstraction leveraged in the previous section is compatible with the *offline replay selection*, which we can therefore leverage to more quickly select the best policy-prompt-LLM combination. To evaluate the statistical performance of selection, we generate 500 unique deployments by randomly permuting 100 trials from a set of 150 distinct maps, expecting the robot to perform selection over all nine policy-prompt-LLM combinations shown in Table 1 separately for each sequence.

In Figure 4, we report the *average navigation cost*, which corresponds to the average of navigation costs incurred in trials 1-through-$k$, averaged over all 500 deployments. The *cumulative regret*, also shown in Figure 4, tracks performance over time as the cumulative difference between the selection-based policy and a Best Performance oracle that knows in advance which strategy is best: LLM+MODEL/P-CONTEXT-B/Gemini. Our results demonstrate a reduction of 6.5% in average cost at the end of 100th trial compared to a standard UCB-bandit selection approach. In particular, we achieve 33.8% lower cumulative regret after 100th trial compared to UCB-bandit selection, a number which would continue to grow with more trials.

Our results highlight the need and benefits of fast deployment-time selection of prompts and LLMs, since without such selection, the robot risks poor performance if it uses only one prompt or LLM preselected before deployment.

Our selection approach enables the robot to quickly pick *during deployment* the prompts and LLMs that yield better behavior and hence maximizing long-term performance—a benefit afforded by our high-level action abstraction amenable to *offline replay* of Paudel & Stein (2023).

| Metric | Selection Approach | Num of Trials ($k$) | | |
|---|---|---|---|---|
| | | $k = 20$ | $k = 50$ | $k = 100$ |
| Avg. Cost | UCB Selection | 258.72▲ | 254.63◆ | 247.25■ |
| | Replay Selection (ours) | **248.12**▲ | **240.40**◆ | **231.29**■ |
| Cumul. Regret | UCB Selection | 884.7△ | 2102.8◇ | 3800.2□ |
| | Replay Selection (ours) | **751.0**△ | **1579.8**◇ | **2516.5**□ |

Figure 4: **Prompt-LLM Selection Results**: Leveraging offline replay for prompt/model selection allows faster selection of the best prompting strategy compared to the UCB, resulting in lower average cost (6.5% improvement) and lower cumulative regret (33.8% improvement).

## 7 REAL-ROBOT DEMONSTRATION

**Policy Performance Results**   We demonstrate the effectiveness of our LLM-informed model-based planning and prompt selection approach with a LoCoBot robot in an apartment containing a kitchen, a dining room, a living room and a bedroom. These rooms contain a total of nine containers where the robot can search for target object as shown in Appendix B. We conduct five trials in which the robot starts in the kitchen and is tasked to find a distinct object in each trial. For the purpose of demonstration, we use GPT-5 as our LLM and use P-CONTEXT-B and P-MINIMAL prompts for LLM+MODEL policy and compare with LLM-DIRECT policy using P-DIRECT prompt. We show one representative example in Figure 5 and report all results in Appendix B. Across five trials, our LLM+MODEL policy outperforms LLM-DIRECT policy by 29% in terms of average navigation cost. See Appendix B for all results and further discussion.

**Prompt Selection Results**   We also deploy our prompt selection approach on the LoCoBot robot with three policy-prompt-LLM combination used in aforementioned object search experiments for five trials in which the robot is tasked to find a distinct object. We observe that our Replay Selection approach quickly selects the best strategy and outperforms UCB bandit by 10.5% in terms of average navigation cost and 34.5% in terms of cumulative regret. See Appendix B for all results and more discussion.

## 8 CONCLUSION

We present a novel framework that seeks to integrate LLMs with high-level model-based reasoning for performant object search. Our approach first introduces an LLM-informed model-based high-level planner for object search in partially-known environments that integrates predictions about

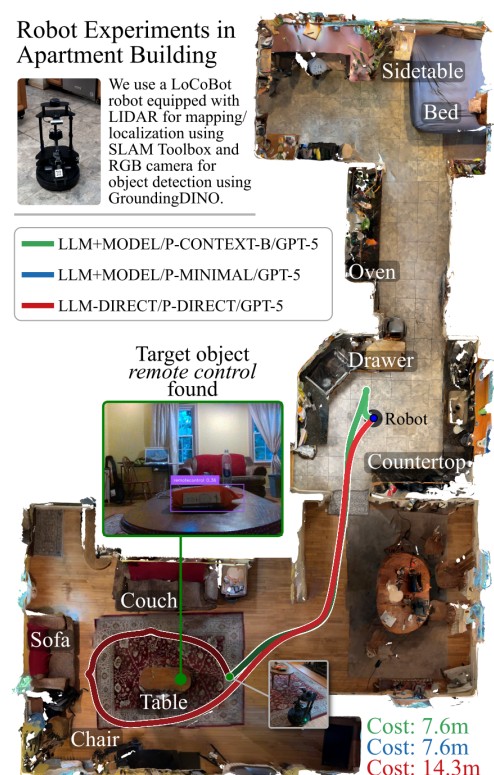

Figure 5: LLM-informed Object Search Experiments in an Apartment

uncertainty from LLMs and information partially-known from the environment. Second, by leveraging the high-level action abstraction upon which model-based planning relies, we demonstrate that the recent *offline replay* approach developed for model selection for learning-informed point-goal navigation (Paudel & Stein, 2023) can be made to support fast deployment-time prompt and LLM selection, a capability unique in this domain. In future, we hope to extend our model-based planning approach to fully unknown environments, which the robot must reveal through exploration and observation. Additionally, our prompts are manually generated and remain unchanged during deployment. In future, we hope to explore automated prompt generation and refinement strategies that integrate with our selection approach~~to allow adaptation to a wide variety of environments during deployment~~. Coupled with our offline replay approach, such methods could enable reliable deployment-time refinement of prompts across diverse environments without requiring on-robot trial-and-error.

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

# APPENDIX

## A SAMPLES OF PROMPTS

As discussed in Section 6, we include below the samples of all prompts used in our experiments.

**P-CONTEXT-A**:

> You are serving as part of a system in which a robot needs to find
> objects located around a household.  Here is a schema that describes
> the connectivity of rooms in the house:  The apartment contains the
> following rooms:  bathroom, bedroom. The bedroom contains:  bed,
> chair, sidetable.  The bathroom contains:  dresser, sidetable, sink,
> toilet.  You will be asked to estimate the probability (a value between
> 1% and 100%) of where objects are located in that house, leveraging
> your considerable experience in how human occupied spaces are located.
> You must produce a numerical value and nothing else, as it is important
> to the overall functioning of the system.  Here is an example exchange
> for an arbitrary house:
> User:  What is the likelihood that I find eggs in the refrigerator in
> the kitchen?
> You:  90%
> The logic here is that there is a high likelihood that a typical
> refrigerator in the kitchen contains eggs, but it is not guaranteed
> as not all refrigerators have eggs.  Here is your prompt for today:
> What is the likelihood that I find book in the sidetable in the
> bedroom?

**Output:** 95%

**P-CONTEXT-B**:

> You are assisting in a robotic system designed to locate items within
> a residence.  The following is a description of the layout and
> connectivity between rooms in the home:  The apartment contains the
> following rooms:  bathroom, bedroom. The bedroom contains:  bed,
> chair, sidetable.  The bathroom contains:  dresser, sidetable, sink,
> toilet.  Your task is to estimate the likelihood (a percentage from
> 1% to 100%) that a specified object is in a given location.  Base your
> reasoning on general patterns of human behavior and usage of household
> spaces.  Your response must be a single numerical value, with no
> additional explanation, as precision is critical to system operation.
> Example exchange:
> User:  What is the probability of finding bread in the pantry in the
> kitchen?
> You:  85%.
> The reasoning here is that bread is commonly stored in pantries, but
> exceptions exist, such as if it is refrigerated.  Now, respond to this
> prompt:
> What is the probability of finding pillow in the bed in the bedroom?

**Output:** 95%

**P-MINIMAL**:

> What is the probability of finding plate in the dining table in the
> kitchen of a typical household?  Your response should only include a
> numerical percentage value between 1% to 100% and nothing else.

**Output:** 80%

**P-DIRECT**:

> You are assisting a robot in locating objects within a household
> based on a provided map of rooms and their contents.  Your task is to
> determine the exact location where the specified object can be found,
> based on given description of the household.  You will be asked pick a
> location to visit where the object could be found quickly.  You should
> only pick one location from the given list.  Here is an example:

```
User:  The apartment contains:  bathroom, bedroom, kitchen.  The
distance between rooms is as follows:  bathroom and bedroom:  5.95
meters, bedroom and kitchen:  3.25 meters, bathroom and kitchen:  4.75
meters.  The robot is currently located at bathroom and is looking for
pillow.  Available locations to search are:  sink in bathroom, toilet
in bathroom, bed in bedroom, sidetable in bedroom.  Which of the given
search locations should the robot visit to find pillow in the least
time?
You:  bed in bedroom
Now give your answer for another household with the following layout:
The apartment contains the following rooms:  bathroom, bedroom.  The
distance between rooms is as follows:  bedroom and bathroom:  5.8
meters.  The robot is currently located at bedroom and is looking
for faucet.  Available locations to search are:  dresser in bathroom,
sidetable in bathroom, sink in bathroom, toilet in bathroom, bed
in bedroom, chair in bedroom, sidetable in bedroom.  Which of the
given search locations should the robot visit to quickly find faucet?
Respond with a search location and nothing else.
```

**Output:** `sink in bathroom`

## B    REAL-ROBOT RESULTS

### B.1    POLICY PERFORMANCE RESULTS

As discussed in Section 7, we report in Table 2 navigation costs of three policy-prompt-LLM combinations for object search in an apartment building. Our LLM+MODEL policy incurs an average navigation cost of 15.1m compared to the baseline LLM-DIRECT policy which incurs the cost of 21.3m, thus outperforming LLM-DIRECT policy by 29%.

| Policy / Prompt / LLM | Target Object | | | | | Average Cost |
|---|---|---|---|---|---|---|
| | *blanket* | *cell phone* | *remote* | *pillow* | *wallet* | |
| LLM+MODEL (ours) / P-CONTEXT-B / GPT-5 | 23.5 | **24.7** | **7.6** | **6.8** | **13.2** | 15.2 |
| LLM+MODEL (ours) / P-MINIMAL / GPT-5 | **23.2** | **24.7** | **7.6** | **6.8** | **13.2** | **15.1** |
| LLM-DIRECT (baseline) / P-DIRECT / GPT-5 | 23.7 | 33.6 | 14.3 | **6.8** | 27.9 | 21.3 |

Table 2: **Real-World LLM-informed Object Search Results:** Our LLM+MODEL policy outperforms baseline LLM-DIRECT policy. All costs are in meters.

### B.2    PROMPT SELECTION RESULTS

As discussed in Section 7, we deploy prompt selection approach onboard a LoCoBot robot with our Replay Selection approach and compare it with UCB Selection approach. The results are shown in Figure 6. Over the course of five trials each with distinct target object, our Replay Selection approach incurs an average cost of 15.2 while UCB Selection incurs an average cost of 17.0m, an improvement of 10.5%. Similarly, our Replay Selection approach incurs a cumulative regret of 21.6 compared to UCB which incurs 33.0, showing an improvement of 34.5%.

## C    DETAILS ABOUT BELIEF UPDATES

As mentioned in Section 5.2, our belief state includes the occupancy map, locations of containers, whether each container has been explored or not, and objects found in each container. Upon exploring a container, the objects found in that container are added to the robot's partial map, and the container is marked as explored. If the target object was not found in that container, planning continues with the updated set of container search actions that only includes unexplored containers, and does not include any container that has been marked as explored. This process is identical for all planning strategies considered in our work.

| Metric | Selection Approach | Num of Trials ($k$) | | |
|---|---|---|---|---|
| | | $k =$ | $k = 3$ | $k = 5$ |
| Avg. Cost | UCB Selection | 28.6▲ | 21.57◆ | 17.0■ |
| | Replay Selection (ours) | **24.1**▲ | **18.6**◆ | **15.2**■ |
| Cumul. Regret | UCB Selection | 21.9△ | 28.3◇ | 33.0□ |
| | Replay Selection (ours) | **17.5**△ | **21.0**◇ | **21.6**□ |

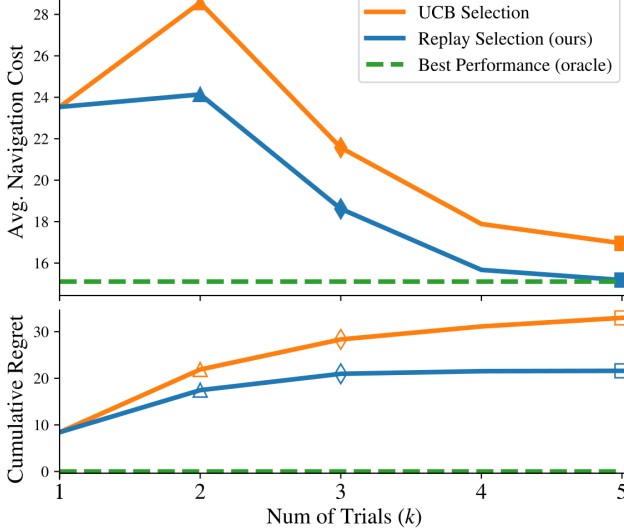

Figure 6: **Prompt-LLM Selection in an Apartment**: Our Replay Selection allows faster selection of the best prompting strategy compared to the UCB selection strategy, resulting in lower average cost and cumulative regret.

## D    SCALABILITY IN TERMS OF NUMBER OF CONTAINERS AND PROMPTS/LLMS

As is true with many exhaustive search-based strategies, the speed of planning scales poorly with the number of containers or apartment size since the number of feasible plans scales factorially with the increase in number of containers. As discussed in Section 4, we limit the action space to speed up planning by using heuristics to choose up to eight containers with high likelihoods $P_S$ and low travel costs $D$ and select among them the container with lowest expected cost, incorporating additional containers as the robot moves and searches. Since the set of actively considered containers is updated as those containers are explored, there is no loss of generality, and this strategy works well in practice and yields effective performance as demonstrated in our experiments.

In terms of the number of prompts and LLMs, our prompt selection approach scales linearly since the addition of a new prompt/LLM would only add the overhead of replaying this new prompt/LLM after a trial is complete. During each replay, the most computational overhead comes from querying the LLM, which we additionally mitigate by caching the LLM's responses from deployment. As such, the replay in itself is quite inexpensive (less than a couple of seconds each), making our prompt/LLM selection approach highly scalable to large number of prompts and LLMs.

## E    USING MARGINAL PROBABILITIES

For each container search action, we obtain likelihood $P_S$ of finding the target object in that container by querying LLMs. These likelihoods are not normalized across available containers because they do not need to be a valid probability mass. For each container, $P_S$ is a marginal probability that represents the likelihood of finding the target object in that container, which is treated as being

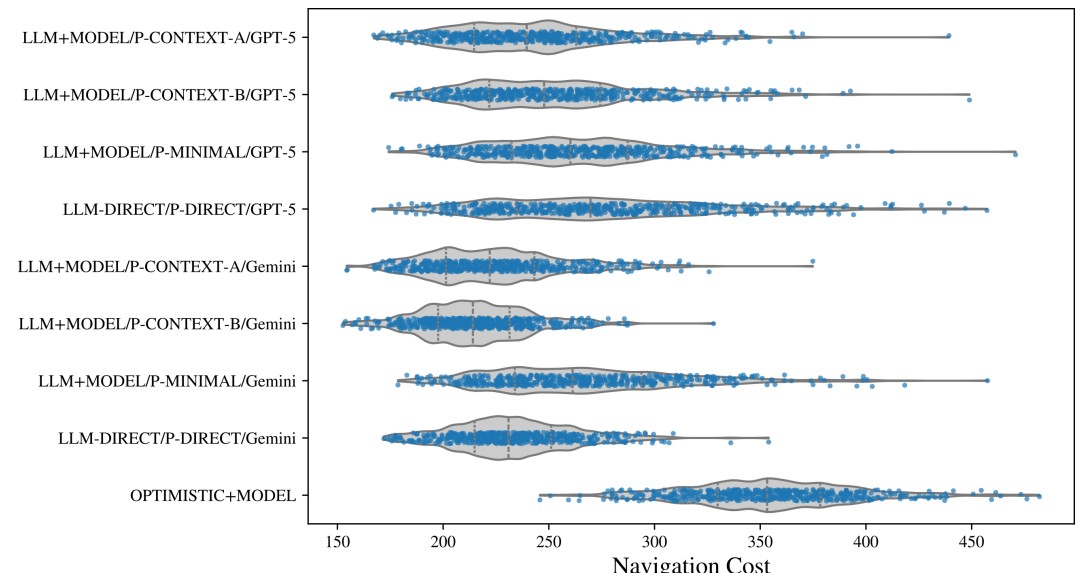

Figure 7: Distribution of navigation costs for different policy/prompt/LLM across 150 PROCTHOR maps. Dotted lines denote quartiles.

| Policy / Prompt / LLM | Avg. Navigation Cost |
|---|---|
| LLM+MODEL (ours) / P-CONTEXT-A / GPT-OSS | **217.12** |
| LLM+MODEL (ours) / P-CONTEXT-B / GPT-OSS | 221.47 |
| LLM+MODEL (ours) / P-MINIMAL / GPT-OSS | 218.78 |
| LLM-DIRECT (baseline) / P-DIRECT / GPT-OSS | 283.60 |
| LLM+MODEL (ours) / P-CONTEXT-A / Llama3.2 | 302.83 |
| LLM+MODEL (ours) / P-CONTEXT-B / Llama3.2 | **281.05** |
| LLM+MODEL (ours) / P-MINIMAL / Llama3.2 | 313.24 |
| LLM-DIRECT (baseline) / P-DIRECT / Llama3.2 | 806.38 |

Table 3: Navigation costs for object search with open-source LLMs: GPT-OSS 120B and Llama3.2 3B. Our LLM+MODEL planners outperform LLM-DIRECT policy that fully relies on LLM.

independent of what other containers exist. The information about what other containers exist and their respective likelihoods are instead used by our planning approach to compute the best action thus alleviating the need to normalize these probabilities across containers. This is a more general formulation of the problem rather than normalizing probabilities across containers which implicitly asserts both that the object must exist and that there is only one to be found—an assumption that may not be valid in the general case.

## F    NAVIGATION COST DISTRIBUTION

In Figure 7, we show violin plots with scatter plots of navigation costs for each policy/prompt/LLM across 150 PROCTHOR maps corresponding to the results in Table 1 for object search experiments.

## G    OBJECT SEARCH EXPERIMENTS WITH OPEN-SOURCE LLMS

In addition to proprietary models, GPT-5 Mini and Gemini 2.5 Flash, we additionally perform object search experiments in 150 PROCTHOR maps with two open-source LLMs: GPT-OSS 120B and Llama3.2 3B. The results are shown in Table 3, which show that our LLM+MODEL planners ourperform baseline LLM-DIRECT policy that fully relies on LLMs for object search, with improvements up to 65.2% for Llama3.2 and 23.4% for GPT-OSS.

| Model | Tokens Used | | | Cost |
| Name | Input | Output | Total | (USD) |
|---|---|---|---|---|
| GPT-5 Mini | 903686 | 9264 | 912950 | $0.14 |
| Gemini 2.5 Flash | 904209 | 8066 | 912275 | $0.24 |

Table 4: Token usage and incurred costs for GPT-5 and Gemini models

## H    TOKEN USAGE AND COSTS

In Table 4, we show approximate number of tokens used and incurred costs to use proprietary models, GPT-5 Mini and Gemini 2.5 Flash, via their respective APIs for our experiments.

