# OpenReview forum: "LLM-informed Object Search in Partially-Known Environments via Model-based Planning and Prompt Selection"
_ICLR.cc/2026/Conference — Submitted to ICLR 2026_

### Official Review · Reviewer_6AAV · 2025-10-30

**Soundness:** 3
**Presentation:** 3
**Contribution:** 2
**Rating:** 4
**Confidence:** 4

**Summary:**

This paper presents an approach to integrate LLMs into planning, in particular for object search in partially observable environments. The key idea is to query LLMs for the probability of an object being in a certain location, and integrate this probability into the Q-Value equation. An LLM/prompt selection strategy is also presented, based on the offline replay approach of Paudel & Stein (2023). Experiments are performed in simulation using procedurally generated homes, and with a real robot in one apartment.

**Strengths:**

- Combining LLM with planning is interesting and relevant.

- Prompt and LLM selection is also studied, adding more technical contributions to the paper.

- The application studied (object search in household environments) is interesting and relevant.

- 150 distinct household environments are used in simulation, and one real apartment is also explored.

- The paper is in general well-written and well-presented.

**Weaknesses:**

- The idea of querying a ML model to bias planning is quite straightforwards. For instance, it reminds me of the AlphaZero approach for on-line planning. The context and technique here is obviously different, but the point is just that the key idea is not very surprising.

- There is no statistical study of the results. Not even the variance is shown.

- Although the paper is well-written, the problem formulation still seems confusing/inconsistent. Sometimes the cost of searching a location seems to be considered, but sometimes only the travelling cost seems to be taken into account.

- It is unclear how this approach would scale, e.g., as the number of containers and apartment size grows. It is also unclear how the approach would scale in terms of potential prompts/LLMS.

- I am quite confused by the prompt selection approach. I would expect it to be used to select prompts and/or LLMs to integrate into the planning approach, but in the end approaches without planning and/or prompts also seem to be considered.

- Baselines could be stronger:

-- The P-DIRECT prompt seems to be ignoring the container search cost R_{search}.

-- OPTIMISTIC+GREEDY seems to also ignore the cost of searching a container.

-- What about directly sampling an action from the probabilities outputted by the LLM?

-- Isn't it possible to compare against existing approaches in the literature? (E.g., PUCT strategy, or some state-of-the-art LLM+planning approach?)

= Detailed Comments =

- "these approaches focus selecting prompts" -> "focus on"

- "Our robot is tasked find a target" -> "to find"

- "without the robot having deploy the plans informed by LLMs" -> "having to"

**Questions:**

1 - Are the results statistically significant? What is the confidence interval of the results?

2 - Could you clarify the problem formulation? Are you considering only the navigation cost, or do you also consider the cost of searching inside a specific container? If the cost of searching a container is considered, why this is later ignored in the paper?

3 - How scalable is the approach to problem size, and prompt/LLM options size?

4 - Are you also considering LLM-DIRECT and OPTIMISTIC+GREEDY as potential options for your prompt selection approach (replay selection), and also for UCB selection? If so, why is that reasonable?

---

> ### Author Response · Authors · 2025-11-25
> **Author Rebuttal to Reviewer 6AAV (Part 1 of 2)**
>
> We thank the reviewer for their thoughtful comments and suggestions. We reply to their individual feedback below:
>
> >**W1:** The idea of querying a ML model to bias planning is quite straightforwards. For instance, it reminds me of the AlphaZero approach for on-line planning. The context and technique here is obviously different, but the point is just that the key idea is not very surprising.
>
> Our primary contributions are twofold: a novel model-based planning framework for object search that integrates a classical planner with commonsense knowledge from LLMs, and a novel approach for deployment-time selection of prompts and LLMs for object search. Although prior works have explored some aspects of these challenges, no existing approach exists that unifies them into a common framework that enables both long-horizon model-based planning and deployment-time introspection for LLM-guided object search.
>
>
> >**W2:** There is no statistical study of the results. Not even the variance is shown.
>
> Our navigation costs are non-Gaussian in nature since the cost incurred by the robot to find an object depends on the map that the robot is deployed in. As such, we reported mean costs in 150 maps for a total of 1350 unique trials across all policies. Instead, in Appendix F of the updated manuscript, we have included violin plots and scatter plots corresponding to each policy/prompt/LLM to show better picture of how each strategy performs in our experiments. These new results further support the conclusion for the paper, validating the effectiveness of our approach.
>
>
> >**W3:** Although the paper is well-written, the problem formulation still seems confusing/inconsistent. Sometimes the cost of searching a location seems to be considered, but sometimes only the travelling cost seems to be taken into account.
>
> Our problem formulation and the proposed planning framework are designed to incorporate search costs (for example, costs associated with opening a fridge or a cabinet), which would be relevant for deployment on a mobile manipulator and easy to include within our framework. For the purposes of our experiments, we do not directly simulate the robot executing manipulation skills, and we treat objects as being instantaneously revealed upon reaching a container and so assign search cost to be zero. We have updated the paper to clarify this point in Sec. 6.1 (1st paragraph).
>
>
> >**W4:** It is unclear how this approach would scale, e.g., as the number of containers and apartment size grows. It is also unclear how the approach would scale in terms of potential prompts/LLMS.
>
> As is true with many exhaustive search-based strategies, the speed of planning indeed can scale poorly with the number of containers or apartment size since the number of feasible plans scales factorially with the increase in number of containers. As such, to speed up planning, we limit the action space using heuristics to choose up to eight containers with high likelihoods (P_S) and low travel costs (D) and select among them the container with lowest expected cost, incorporating additional containers as the robot moves and searches. Since the set of actively considered containers is updated as those containers are explored, there is no loss of generality, and this strategy works well in practice yielding effective performance as demonstrated in our experiments. We have included these clarifications in Sec. 4 and Appendix D of the updated manuscript.
>
> In terms of the number of prompts/LLMs, our prompt selection approach scales linearly since the addition of a new prompt/LLM would only add the overhead of replaying this new prompt/LLM after a trial is complete. During each replay, the most computational overhead comes from querying the LLM, which we additionally mitigate by caching the LLM responses from deployment. As such, the replay in itself is quite inexpensive (less than a couple of seconds each), making our prompt/LLM selection approach highly scalable to large number of prompts/LLMs. We have included this discussion in Appendix D of the updated manuscript.
>
> >**W5:** I am quite confused by the prompt selection approach. I would expect it to be used to select prompts and/or LLMs to integrate into the planning approach, but in the end approaches without planning and/or prompts also seem to be considered.
>
> As discussed in L83-89 of the original manuscript, our LLM-informed model-based planning approach and our prompt/LLM selection approach are collectively enabled by our high-level action abstraction, as opposed to the prompt/LLM selection approach being enabled by the planner itself. As such, our other strategies that do not explicitly use planning or query LLMs—yet use the same underlying abstraction—can be integrated into our prompt/LLM selection approach, enabling much broader applications. We include this updated discussion in Sec. 6.1 (Prompt Selection) of the updated manuscript.
>
> [continued below...]

---

> ### Author Response · Authors · 2025-11-25
> **Author Rebuttal to Reviewer 6AAV (Part 2 of 2)**
>
> [...continued from above]
>
> >**W6.1,W6.2:** Baselines could be stronger: -- The P-DIRECT prompt seems to be ignoring the container search cost R_{search}. -- OPTIMISTIC+GREEDY seems to also ignore the cost of searching a container.
>
> As addressed in response to Weakness 3, we assume container search cost to be zero since our robot does not have manipulation skills. As such, we ignore the search cost for all strategies (ours and baselines) in our experiments, putting all approaches on a level playing field. This has been clarified in Sec. 6.1 of the updated manuscript.
>
> >**W6.3,6.4:** -- What about directly sampling an action from the probabilities outputted by the LLM?
> -- Isn't it possible to compare against existing approaches in the literature? (E.g., PUCT strategy, or some state-of-the-art LLM+planning approach?)
>
> Our primary contributions are an LLM-informed model-based planning framework and a deployment-time prompt/LLM selection approach that are enabled by a high-level abstraction for object search problem. While additional baselines would have been interesting to explore, we believe that our current experiments and results sufficiently validate our proposed contributions. Additionally, adding these existing approaches as a part of candidate strategies for our deployment-time selection as afforded by our prompt/LLM selection approach would be an interesting future work to explore.
>
> >**W7:** = Detailed Comments =
> >   - "these approaches focus selecting prompts" -> "focus on"
> >   - "Our robot is tasked find a target" -> "to find"
> >   - "without the robot having deploy the plans informed by LLMs" -> "having to"
>
> We have updated the manuscript to fix these errors.
>
> >**Q1:** Are the results statistically significant? What is the confidence interval of the results?
>
> Please see our response to W2 where we have addressed this concern.
>
> >**Q2:** Could you clarify the problem formulation? Are you considering only the navigation cost, or do you also consider the cost of searching inside a specific container? If the cost of searching a container is considered, why this is later ignored in the paper?
>
> Please see our response to W3 where we have addressed this concern.
>
> >**Q3:** How scalable is the approach to problem size, and prompt/LLM options size?
>
> Please see our response to W4 where we have addressed this concern.
>
> >**Q4:** Are you also considering LLM-DIRECT and OPTIMISTIC+GREEDY as potential options for your prompt selection approach (replay selection), and also for UCB selection? If so, why is that reasonable?
>
> Our prompt selection approach relies on offline replay, which can replay the robot behavior of not just alternative prompts and LLMs, but also other strategies that use the same underlying high-level action abstraction to determine the best performing among all strategies. As such, the LLM-DIRECT and OPTIMISTIC+GREEDY strategies, which use the same high-level action abstraction as other strategies, are also considered for our deployment-time selection approach. We have updated our manuscript in Sec. 6.1 (Prompt Selection) to clarify this further.

---

### Official Review · Reviewer_WQ43 · 2025-10-30

**Soundness:** 3
**Presentation:** 3
**Contribution:** 3
**Rating:** 6
**Confidence:** 4

**Summary:**

This paper presents a framework for LLM-informed object search in partially-known household environments, tackling the challenge of long-horizon planning under uncertainty. The core contribution is the LLM+MODEL planning approach, which uses a Large Language Model (LLM) not to directly dictate actions, but to provide an estimate of the object finding likelihood, $P_S$. This likelihood is integrated into a Bellman-like equation that minimizes the expected navigation cost. The work further introduces a method for fast deployment-time selection of the best prompt/LLM combination, leveraging an offline replay mechanism compatible with the high-level action abstraction. Experiments in simulation (ProcTHOR) and on a real robot (LoCoBot) show that the LLM+MODEL planner significantly outperforms LLM-DIRECT and purely optimistic baselines, and the replay selection outperforms standard UCB-bandit selection.

**Strengths:**

The paper presents an interesting approach by integrating Large Language Models (LLMs) with formal model-based planning for long-horizon object search, a strategy that intelligently mitigates the LLM's known shortcomings in quantitative planning.

Instead of using the LLM as a brittle planner, the work uses it as a robust knowledge source to estimate the object finding likelihood, $P_S$, and incorporates this into a planning equation. This clever formulation enables reasoning about the future consequences of actions, leading to significantly better performance than both LLM-direct and myopic greedy baselines.

I think that this approach can be highly practical due to the novel Replay Selection mechanism, which leverages the planning framework to quickly identify the best-performing LLM/prompt combination during deployment. The quality of the validation is commendable, with experiments conducted across two distinct state-of-the-art LLMs (GPT-5 and Gemini 2.5) in large-scale simulation and successfully verified on a real robot (LoCoBot).

**Weaknesses:**

A few weaknesses:

1. The probability $P_S$ is generated by prompting the LLM to output a numerical percentage value (e.g., "95%"). It is unclear how this raw, subjective LLM output is interpreted or normalized to be a valid probability mass, $\sum_{i} P_S(a_i) \leq 1$, across all available containers $a_i$. Simply asking for a probability does not guarantee a statistically or mathematically rigorous distribution across all containers.
2. While the concept of offline replay is compelling, the description (Section 5.3) states that for an alternative policy $\pi_{\theta^{\prime}}$, the cost is computed by "pessimistically assuming that all other containers would not have contained the target object" This seems like a strong assumption that might bias the replayed cost. It is not entirely clear if the replayed policy $\pi_{\theta^{\prime}}$ re-plans its actions at every step based on its "simulated" belief state, $b_t'$, or if it simulates a full, fixed sequence of actions generated at $t=0$. It would be great if the authors could provide further clarification.
3. The LLM’s role is restricted to outputting a single number ($P_S$). This is a very narrow use of the LLM's reasoning capability. The action space is also restricted to searching pre-defined containers. This limits the robot’s ability to plan novel exploration strategies or decide if a room itself is worth exploring, which is critical in a fully unknown environment, a stated goal for future work. But I am confused with the motivation for using the LLMs in this manner. Could the authors provide more clarification on this?
4. The experiments does not include comparisons with open-source LLMs (only Gemini and GPT-5 are included). Also, the authors should provide approximate tokens used for their experiments and the costs associated with querying close sourced models.

Few missing reference that are relevant to this topic and would be a good addition:
1. Zhang, X., Qin, H., Wang, F., Dong, Y., & Li, J. (2025, May). Lamma-p: Generalizable multi-agent long-horizon task allocation and planning with lm-driven pddl planner. In 2025 IEEE International Conference on Robotics and Automation (ICRA) (pp. 10221-10221). IEEE.
2. Nayak, S., Morrison Orozco, A., Have, M., Zhang, J., Thirumalai, V., Chen, D., ... & Balakrishnan, H. (2024). Long-horizon planning for multi-agent robots in partially observable environments. Advances in Neural Information Processing Systems, 37, 67929-67967.
3. Ling, S., Wang, Y., Fan, C., Lam, T. L., & Hu, J. (2025). ELHPlan: Efficient Long-Horizon Task Planning for Multi-Agent Collaboration https://www.arxiv.org/abs/2509.24230

**Questions:**

1. Please clarify how the individual likelihood values $P_S(a_t)$ provided by the LLM are used in the Bellman equation (Eq. 3). If the LLM returns $90$% for container $A$ and $80$% for container $B$ (both possible from the prompt design), how do you ensure that the search actions $a_A$ and $a_B$ respect the laws of probability in the model? Is there a normalization step applied to the raw LLM outputs $\hat{P}_S$ such that $\sum_{a_i \in \mathcal{A}(b_t)} P_S(a_i) \leq 1$?
2. The OPTIMISTIC+GREEDY baseline is uninformed. A more informative baseline would be a model-based planner that uses a naive, uniform probability distribution for $P_S$ (i.e., $P_S = 1/|\mathcal{A}(b_t)|$ for all containers) or perhaps one based on environment size, rather than LLM commonsense. Did the authors compare against a non-LLM, purely Information-Theoretic Planning baseline? This would quantify the *true* value of the LLM's commonsense beyond simply outperforming a greedy approach. Another baseline to consider would be LLaMAR [1] where they use a heuristic-based exploration to find objects relevant to the search using semantic matching through a sentence transformer.
3. Table 1 shows that for Gemini, P-CONTEXT-B performs best, while for GPT-5, P-CONTEXT-A performs best. Since the semantic difference between P-CONTEXT-A and P-CONTEXT-B is minimal (”differ in terms of the language is used in the prompt text” ), could the authors elaborate on *why* this subtle linguistic difference causes such a significant performance divergence between the two LLMs? This insight would be valuable for the LLM community. I believe that this leads us to the ongoing research question of how much does the prompt affect the performance of LLMs for different tasks.
4. Currently, the action space $\mathcal{A}$ consists of pre-identified containers. Could the high-level action abstraction be redefined to also include an action, $a_{\text{explore}}$, where the robot navigates to a new room/area? This would be key to extending the work to the “fully unknown environments” mentioned in the conclusion.

[1]: Nayak, S., Morrison Orozco, A., Have, M., Zhang, J., Thirumalai, V., Chen, D., ... & Balakrishnan, H. (2024). Long-horizon planning for multi-agent robots in partially observable environments. Advances in Neural Information Processing Systems, 37, 67929-67967.

---

> ### Author Response · Authors · 2025-11-25
> **Author Rebuttal to Reviewer WQ43 (Part 1 of 2)**
>
> We thank the reviewer for their thoughtful comments and suggestions. We reply to their individual feedback below:
>
> >**W1:** The probability $P_S$ is generated by prompting the LLM to output a numerical percentage value (e.g., "95%"). It is unclear how this raw, subjective LLM output is interpreted or normalized to be a valid probability mass, $\sum_{i} P_S(a_i) \leq 1$, across all available containers $a_i$. Simply asking for a probability does not guarantee a statistically or mathematically rigorous distribution across all containers.
>
> We would like to clarify that we do not normalize the probabilities obtained from LLMs, and they do not need to be a valid probability mass across available containers. For each container, $P_S$ is a marginal probability that represents the likelihood of finding the target object in that container, which is treated as being independent of what other containers exist. The information about what other containers exist and their respective likelihoods are instead used by our planning approach to compute the best action thus alleviating the need to normalize these probabilities across containers. This is a more general formulation of the problem rather that normalizing probabilities across containers which implicitly asserts both that the object must exist and that there is only one to be found—an assumption that may not be valid in the general case. We have added this discussion in Appendix E of the updated manuscript.
>
> >**W2:** While the concept of offline replay is compelling, the description (Section 5.3) states that for an alternative policy $\pi_{\theta^{\prime}}$, the cost is computed by "pessimistically assuming that all other containers would not have contained the target object" This seems like a strong assumption that might bias the replayed cost. It is not entirely clear if the replayed policy $\pi_{\theta^{\prime}}$ re-plans its actions at every step based on its "simulated" belief state, $b_t'$, or if it simulates a full, fixed sequence of actions generated at $t=0$. It would be great if the authors could provide further clarification.
>
> Regarding the pessimistic assumption, we agree that this might bias the replayed cost in cases when more than one instance of target object may be available in the environment. However, this assumption has been empirically shown to constrain bandit-like selection more tightly than other relaxed assumptions in prior work [1] that inspired our prompt replay approach. Given the results of our prompt selection approach based on this assumption which further validates its effectiveness, we believe this to be reasonable assumption. We have added this clarification in Sec. 5.3 of the updated manuscript.
>
> The replayed policy emulates the actual deployment except that the observations upon executing a search action are simulated using data collected from deployment. As such, the replayed policy replans at every step until it “reaches” the container with target object. We have updated the language in Sec. 5.3 to make this clearer in the updated manuscript.
>
> [1] Paudel, A., and Stein G. J.. Data-Efficient Policy Selection for Navigation in Partial Maps via Subgoal-Based Abstraction. IROS 2023.
>
> >**W3:** The LLM’s role is restricted to outputting a single number ($P_S$). This is a very narrow use of the LLM's reasoning capability. The action space is also restricted to searching pre-defined containers. This limits the robot’s ability to plan novel exploration strategies or decide if a room itself is worth exploring, which is critical in a fully unknown environment, a stated goal for future work. But I am confused with the motivation for using the LLMs in this manner. Could the authors provide more clarification on this?
>
> One of the motivations of our work is to leverage what we know already know or can infer based on available knowledge, and use LLMs to fill in missing pieces of the problem. As such, we use LLMs only to get probability values (or to get container search action to execute) since other information required for planning does not need to be obtained from LLMs. We believe this is to be a reasonable and responsible use of LLMs for the problem we have addressed in our work. Regarding abilities for planning novel strategies, we’d like to clarify that our proposed planning framework is quite general and supports different high-level actions. As such, while we instantiate this framework by considering high-level actions corresponding to searching pre-defined containers, the framework itself can support actions that may corresponding to novel exploration strategies or even whether to explore a room. In such scenarios, the use of LLMs/VLMs may be broader than just getting probabilities, and may be useful for broader purposes, e.g. generate candidate exploratory actions in fully unknown environments.
>
> [continued below...]

---

> ### Author Response · Authors · 2025-11-25
> **Author Rebuttal to Reviewer WQ43 (Part 2 of 2)**
>
> [...continued from above]
>
> >**W4:** The experiments does not include comparisons with open-source LLMs (only Gemini and GPT-5 are included). Also, the authors should provide approximate tokens used for their experiments and the costs associated with querying close sourced models.
>
> While we only experiment with Gemini and GPT-5, the primary contribution of the our paper, namely a model-based planning framework and a prompt selection approach, are agnostic to whether we use closed-source or open-source models. We use close-source models for their convenience to setup and integrate. However, we appreciate your concern, and have additional experiments in Appendix G with Llama3.2 3B and GPT-OSS 120B open source models. Consistent with existing results, our LLM-informed model-based planning approach outperforms those that fully rely on LLMs, with improvements up to 65.2% for Llama3.2 and 23.4% for GPT-OSS.
>
> Additionally, we have updated our manuscript to include approximate tokens used and costs incurred for GPT-5 and Gemini in Appendix H.
>
> >**W5:** Few missing reference that are relevant to this topic and would be a good addition...
>
> We have updated our paper to include these references.
>
> >**Q1:** Please clarify how the individual likelihood values $P_S(a_t)$ provided by the LLM are used in the Bellman equation (Eq. 3). If the LLM returns $90$% for container $A$ and $80$% for container $B$ (both possible from the prompt design), how do you ensure that the search actions $a_A$ and $a_B$ respect the laws of probability in the model? Is there a normalization step applied to the raw LLM outputs $\hat{P}S$ such that $\sum{a_i \in \mathcal{A}(b_t)} P_S(a_i) \leq 1$?
>
> Please see our comment above in response to W1 where we have discussed this.
>
> >**Q2:** The OPTIMISTIC+GREEDY baseline is uninformed. A more informative baseline would be a model-based planner that uses a naive, uniform probability distribution for $P_S$ (i.e., $P_S = 1/|\mathcal{A}(b_t)|$ for all containers) or perhaps one based on environment size, rather than LLM commonsense. Did the authors compare against a non-LLM, purely Information-Theoretic Planning baseline? This would quantify the true value of the LLM's commonsense beyond simply outperforming a greedy approach. Another baseline to consider would be LLaMAR [1] where they use a heuristic-based exploration to find objects relevant to the search using semantic matching through a sentence transformer.
>
> We would like to clarify that our OPTIMISTIC+GREEDY assumes P_S=1 for all containers, and hence the model-based planning would be equivalent to picking the closest unexplored container (i.e. OPTIMISTIC+GREEDY) since minimization would effectively be over traversal costs to unexplored containers.
>
> With regards to other baselines, we would like to reiterate that our primary contributions are a model-based framework that integrates of planning and LLMs, and how our abstraction enable us to perform deployment-time prompt/LLM selection. While it would be interesting to experiment with other LLM or non-LLM baselines, we believe that our current experiments sufficiently demonstrate the value of our integrated model-based planning framework and prompt selection approach—the core contributions of our work.
>
> >**Q3:** Table 1 shows that for Gemini, P-CONTEXT-B performs best, while for GPT-5, P-CONTEXT-A performs best. Since the semantic difference between P-CONTEXT-A and P-CONTEXT-B is minimal (”differ in terms of the language is used in the prompt text” ), could the authors elaborate on why this subtle linguistic difference causes such a significant performance divergence between the two LLMs? This insight would be valuable for the LLM community. I believe that this leads us to the ongoing research question of how much does the prompt affect the performance of LLMs for different tasks.
>
> It is certainly interesting that the performance diverges across LLMs in response to minimal semantic differences in prompts. This result underscores the importance of our deployment-time prompt selection method, since these sensitivities are difficult to anticipate beforehand. However, we do not attempt to explain the underlying cause of this discrepancy. Doing so would require tools and analyses aimed at understanding the internal mechanics of LLMs—tools that fall outside the scope of our framework, which treats LLMs as black-box models. While deeper investigation would be valuable, it would constitute a separate line of work.
>
> >**Q4:** Currently, the action space $\mathcal{A}$ consists of pre-identified containers. Could the high-level action abstraction be redefined to also include an action, $a_{\text{explore}}$, where the robot navigates to a new room/area? This would be key to extending the work to the “fully unknown environments” mentioned in the conclusion.
>
> Please see our comment above in response to W3 where we have discussed this.

---

### Official Review · Reviewer_yYRt · 2025-10-31

**Soundness:** 3
**Presentation:** 2
**Contribution:** 2
**Rating:** 2
**Confidence:** 3

**Summary:**

This paper addresses two key challenges in object search within partially known environments using LLM-based planning: (1) how to leverage an LLM’s commonsense world knowledge to reduce the cost of finding a target object, and (2) how to select the optimal prompt–LLM combination in an online setting. To tackle the first challenge, the paper introduces a Bellman-style function that computes the expected action cost while incorporating the LLM’s knowledge about how likely a target object is to appear in a given location, based on the agent’s current belief. For the second challenge, the authors employ a replay buffer that stores completed search runs—including ground-truth object locations—to retrospectively evaluate the performance of different prompt–LLM combinations (“what would have happened” if a given combination had been used). Using these estimated performances, a UCB-style algorithm dynamically selects the most promising combination online. Through both simulation and real-robot experiments, the paper demonstrates consistent improvements over baseline methods.

**Strengths:**

Quality: this paper set up the problem clearly and easy to follow. one plus is the paper not only provides simulation experimental results but also show demonstrated improvements in real robot experiments.

**Weaknesses:**

- When introducing the Bellman equation, the authors could provide a more detailed explanation of how the Q-function is computed through dynamic programming. A step-by-step illustration of the recursive computation process would clarify how expected costs are propagated across belief states.
- My main concern of this paper is the formulation of the second problem—selecting the best prompt and LLM combination—is arguably limited in its general applicability to LLM-based object search. For prompt optimization, it would be more natural to use language feedback to iteratively refine prompts, rather than relying on a fixed, predefined set of prompts and applying a bandit-style selection algorithm. As implemented, the best achievable prompt is constrained by the initial candidate set, which may restrict the system’s adaptability and overall performance. Maybe the author can explain why you select this setting.

**Questions:**

1. How is the belief state updated when the robot not found the target location in the containers it currently is searching?
2. In the prompt design, LLM doesn't have access to previous searching location of the robots, which potentially could speed up the searching. Without this information given to LLM, how LLM differs its output from different time step or different known information?

---

> ### Author Response · Authors · 2025-11-25
> **Author Rebuttal to Reviewer yYRt**
>
> We thank the reviewer for their thoughtful comments and suggestions. We reply to their individual feedback below:
> >**W1:** When introducing the Bellman equation, the authors could provide a more detailed explanation of how the Q-function is computed through dynamic programming. A step-by-step illustration of the recursive computation process would clarify how expected costs are propagated across belief states.
>
> We appreciate the reviewer’s suggestion for a detailed illustration of expected cost computation. Fig. 1(a) includes an annotated example of a plan from our approach, which represents a single trace from the search tree. We have updated the figure caption to highlight the relationship between Fig. 1(a) and the search over policies.
>
> >**W2:** My main concern of this paper is the formulation of the second problem—selecting the best prompt and LLM combination—is arguably limited in its general applicability to LLM-based object search. For prompt optimization, it would be more natural to use language feedback to iteratively refine prompts, rather than relying on a fixed, predefined set of prompts and applying a bandit-style selection algorithm. As implemented, the best achievable prompt is constrained by the initial candidate set, which may restrict the system’s adaptability and overall performance. Maybe the author can explain why you select this setting.
>
> Our approach for selecting the best prompt and LLM is not limited to LLM-based object search, especially since our approach in itself is inspired by prior work [1] as discussed in the paper, which dealt with point-goal navigation in fully unknown environments and did not rely on LLMs.
>
> To address your concern regarding prompt optimization, we emphasize that a key challenge during deployment is determining whether a newly refined prompt would actually improve the robot’s performance, since untested prompts cannot be reliably evaluated on board. The offline replay approach addresses this challenge by allowing candidate prompts to be evaluated before deployment, ensuring reliable performance. In addition, our method selects not only between prompts but also between LLMs, making it more general than approaches that focus solely on prompt optimization. While we agree that iteratively refining prompts based on language feedback is a natural extension—which we already highlight as potential future work—our current contributions provide a practical foundation for reliable deployment of LLM-guided robots, with future work able to explore deployment-time prompt refinement by building upon this work. We have updated our future work discussion in Sec. 8 to include some of this discussion.
>
> [1] Paudel, A., and Stein G. J.; Data-Efficient Policy Selection for Navigation in Partial Maps via Subgoal-Based Abstraction. IROS 2023.
>
> >**Q1:** How is the belief state updated when the robot not found the target location in the containers it currently is searching?
>
> Our belief state includes the occupancy map, locations of containers, whether each container has been explored or not, and objects found in each container. Upon exploring a container, the objects found in that container are added to the robot’s partial map, and the container is marked as explored. If the target object was not found in that container, planning continues with the updated set of container search actions that only includes unexplored containers, and does not include any container that has been marked as explored. This process is identical for all planning strategies considered in our work. In addition to the existing discussion regarding this in Sec. 4 and 5.2 of the original manuscript, we have further clarified these details in Sec. 4 and Appendix C of the updated manuscript.
>
> >**Q2:** In the prompt design, LLM doesn't have access to previous searching location of the robots, which potentially could speed up the searching. Without this information given to LLM, how LLM differs its output from different time step or different known information?
>
> As mentioned in the previous response, we update the partial map to keep track of unexplored containers, which is then used to determine the set of container search actions. As the robot replans upon not finding the target object and marking the container as explored, the old container search actions are not longer a part of the robot’s candidate action set. This applies to all planning strategies considered in our work. In addition to the existing discussion regarding this in Sec. 4 and 5.2 of the original manuscript, we have further clarified these details in Sec. 4 and Appendix C of the updated manuscript.

---

### Official Review · Reviewer_drFT · 2025-11-01

**Soundness:** 3
**Presentation:** 2
**Contribution:** 2
**Rating:** 4
**Confidence:** 3

**Summary:**

The paper proposes using a zero-shot probability generated from an LLM to contribute to calculating the cost of finding a target object in a candidate container. They also incorporate distance and search costs to the overall cost. This approach allows for increased efficiency and performance in finding target objects. They also contribute an offline replay approach to tune prompt and LLM selection. They provide multiple variations of their methods and provide real-robotic results to show improvement over UCB.

**Strengths:**

-The aspect of automatically finding the best LLM for the task is an important area of research that is under explored, and I appreciate it being tackled in this work

-I like how the authors build upon previous zero-shot LLM approaches for object navigation and also use distance costs to inform decisions

-Authors provide multiple variations of their approach

-Strong results with real-world robot demonstrations

**Weaknesses:**

-L47, some citations are necessary to backup the claim of poor performance on quantitative reasoning tasks, especially since L46 mentions “it is well-established”.

-In Figure 1, why is P_S “informed by LLM” as annotated underneath the equation, when it is the probability of finding the object at the given action/container?  The probability that a book is in a container is independent of what the LLM thinks, so why is the probability based on the LLM, or maybe I’m misunderstanding what the authors mean by “informed by LLM”. Maybe there should a \hat{P}_S that is “informed by LLM” and then during checking P_S is the true probability.

**Questions:**

Minor comments
-In L39, “long term goodness” is a bit vague, perhaps “long term feasibility” or “long term optimality”. Or “long term value” to align more with sequential planning/RL?

-In L50, could the authors give specific ambiguities or design questions that need to considered when integrate LLMs with model-based planners? That will help will clarifying the motivation. When I first read it, I assumed the authors meant prompting strategies to encode information of the environment model to the LLM, but then the next paragraph talks about prompting strategies “In addition”, so I am unclear what the authors refer to in L50.

---

> ### Author Response · Authors · 2025-11-25
> **Author Rebuttal to Reviewer drFT**
>
> We thank the reviewer for their thoughtful comments and suggestions. We reply to their individual feedback below:
> > **W1:** L47, some citations are necessary to backup the claim of poor performance on quantitative reasoning tasks, especially since L46 mentions “it is well-established”.
>
> Thank you for pointing it out. We have added the necessary citations to back this claim.
>
> > **W2:** In Figure 1, why is P_S “informed by LLM” as annotated underneath the equation, when it is the probability of finding the object at the given action/container? The probability that a book is in a container is independent of what the LLM thinks, so why is the probability based on the LLM, or maybe I’m misunderstanding what the authors mean by “informed by LLM”. Maybe there should a \hat{P}_S that is “informed by LLM” and then during checking P_S is the true probability.
>
> We agree that the LLM indeed cannot know precisely where the book is located, though we rely on the fact that the commonsense world knowledge captured by the LLM can still be used to make reasonable predictions about where objects of interest are likely to be, a fact corroborated by the effective object search performance of our approach compared to baselines. In the equation in Fig. 1, $P_S$ denotes the underlying “ground truth” probability, which is not known in general. We label it as “informed by LLM” to mean that an estimated value provided by LLMs is used as an approximation instead. For clarity, we have updated the figure to instead read “estimated using LLM”.
>
> > **Q1:** Minor comments -In L39, “long term goodness” is a bit vague, perhaps “long term feasibility” or “long term optimality”. Or “long term value” to align more with sequential planning/RL?
>
> We have updated the paper to say “long-term value” for more clarity.
>
> >**Q2:** In L50, could the authors give specific ambiguities or design questions that need to considered when integrate LLMs with model-based planners? That will help will clarifying the motivation. When I first read it, I assumed the authors meant prompting strategies to encode information of the environment model to the LLM, but then the next paragraph talks about prompting strategies “In addition”, so I am unclear what the authors refer to in L50.
>
> A key motivation of our approach is to integrate the power of structured model-based planning with the commonsense world knowledge from LLMs. In the space of emerging trends where planning is often offloaded fully to LLMs, our approach aims to integrate a dedicated planner with information obtained from LLMs. One key design question for such an approach is what part of the problem should be handled by LLMs and what part should be left for the planner. Our proposed approach with a high-level action abstraction is an effective answer to this design choice: stochastic transition probabilities for object search actions are hard to model directly and hence LLMs (which we use to approximate these transition probabilities) serve as proxy of commonsense world knowledge. Our model-based planning framework then uses the partial map of the environment to compute the set of container search actions and their associated traversal/search costs without the need to rely on the LLM. In our manuscript, we demonstrate effective performance under uncertainty leveraging such integration of planning and LLMs.
>
> Regarding the prompting strategies, we would like to clarify that we use prompting strategy as a broad term to include choices of text used to query the LLMs (which includes, among others, the information about the robot’s knowledge of the environment, in-context examples, etc.). We have updated our manuscript in Sec. 1 (3rd paragraph) to communicate this more clearly.

---

> > ### Comment · Reviewer_drFT · 2025-11-26
> >
> > Thank you for the response. I appreciate the incorporation of the suggestions in the paper. I am happy to raise to score.

---

### Author Response · Authors · 2025-11-25
**Meta-Rebuttal by Authors**

We thank the reviewers and area chair for their time, patience, and constructive feedback. We are glad to see the novelty, value, and impact of our contributions highlighted by the reviewers. We have responded individually to each of the reviewer comments and look forward to constructive discussion during the rebuttal phase. In line with our responses to reviewers, we have also uploaded the revised manuscript, which includes new experiments, results and analyses, and added/updated discussion—all highlighted in blue. Below, we summarize the key changes we have made in the revised manuscript to address the reviewer comments:

- Clarified language in multiple areas throughout the paper to better communicate and highlight our contributions.
- Added eight new object search batch experiments and results with open-source LLMs: GPT-OSS and Llama3.2, included in a new appendix.
- Added and updated discussion regarding implementation and theoretical insights, including heuristics for planing efficiency, belief updates, offline replay and pessimistic assumptions, costs for searching (R_search), that the estimated terms represent marginal probabilities, prompt selection over all strategies, the potential for future work involving iterative prompt refinement, and scalability over containers/apartment size and prompts/LLMs, among others. These updates appear in main text, appendix or both.
- Added new violin/scatter plots to better illustrate distribution of navigation costs from our simulation experiments for better insight into the statistics and distribution of costs among different object search strategies.
- Provided information about token use and incurred costs for proprietary LLMs used in our experiments.
- Included additional relevant references based on reviewer feedback.

---

### Meta-Review · Area_Chair_Fshw · 2026-01-09

**Summary:**

The paper is using commonsense priors in LLMs to solve object search. The paper has 2 contributions
1. Use the LLM priors to provide a cost that guides search towards more likely spots through a Bellman-esque backup.
2. Optimize prompts through offline replay on deployment.

The method is used with closed source LLMs and applied to sim and real object search problems.

Reviews did recognize the merits of the paper but had some concerns.
1. “LLM probability” vs “true probability” - its not clear if the LLM has a proper probability or some unnormalized score.
2. Prompt optimization felt limited in scope to using language feedback more open-world.
3. Pessimistic replay may bias results
4. More rigorous and performant baselines.
5. Overall felt the prompt optimization piece was hard to understand and the whole mechanism felt a bit ad-hoc.

**Reviewer Concerns:**

The rebuttal ended up clarifying some of the notational questions and improved some notational/discussion clarity. They also added results with open source LLMs.

The major questions about the prompt optimization section were still chalked down to empiricism or not really answered. I think this part of the paper is still not quite ready for publication.

**Reviewer Scores:**

drFT would have upped their score, but I think yYRt, WQ43, 6AAV would not have gone up beyond a 5 or so. I don't think the rebuttal addressed their concerns in enough detail to merit a full score change.

---

### Decision · Program_Chairs · 2026-01-26

Reject